# Dendrobine Alleviates Cellular Senescence and Osteoarthritis via the ROS/NF-κB Axis

**DOI:** 10.3390/ijms24032365

**Published:** 2023-01-25

**Authors:** Haitao Chen, Ming Tu, Siyi Liu, Yinxian Wen, Liaobin Chen

**Affiliations:** Division of Joint Surgery and Sports Medicine, Department of Orthopedic Surgery, Zhongnan Hospital of Wuhan University, Wuhan 430071, China

**Keywords:** osteoarthritis, dendrobine, chondrocyte senescence, reactive oxygen species, NF-κB pathway

## Abstract

Osteoarthritis (OA) is a degenerative joint disease characterized by low-grade inflammation and cartilage degradation. Dendrobine (DEN) is reported to inhibit inflammation and oxidative stress in some diseases, but its role in chondrocyte senescence and OA progress has not yet been elucidated. Our study aimed to explore the protective effects of DEN on OA both in vitro and in vivo. We found that DEN inhibited extracellular matrix (ECM) degradation and promoted ECM synthesis. Meanwhile, DEN inhibited senescence-associated secretory phenotype (SASP) factors expression and senescence phenotype in IL-1β-treated chondrocytes. Furthermore, DEN improved mitochondrial function and reduced the production of intracellular reactive oxygen species (ROS). Also, DEN suppressed IL-1β-induced activation of the NF-κB pathway. Further, using NAC (ROS inhibitor), we found that DEN might inhibit NF-κB cascades by reducing ROS. Additionally, X-ray, micro-CT, and histological analyses in vivo demonstrated that DEN significantly alleviated cartilage inflammation, ECM degradation, and subchondral alterations in OA progression. In conclusion, DEN inhibits SASP factors expression and senescence phenotype in chondrocytes and alleviated the progression of OA via the ROS/NF-κB axis, which provides innovative strategies for the treatment of OA.

## 1. Introduction

Osteoarthritis (OA) is the most common form of joint disease characterized by cartilage degeneration. The progress of OA eventually leads to the disability of elderly patients, which brings a huge burden to the social economy [1,2]. The pathogenesis of OA seems to be a complex interplay of genetic, biochemical, and biomechanical factors [3,4]. Accumulating evidence indicates that low-grade inflammation carries out a key role in the occurrence and development of OA [5,6]. Meanwhile, several lines of investigation have shown that oxidative stress can promote the development of OA [7,8]. Therefore, inhibiting the inflammation and oxidative stress of the synovial joint may be an effective choice for the treatment of OA.

Cellular senescence is usually defined as permanent cell cycle arrest in response to replicative stress and aging [9]. Increased expression of cyclin-dependent kinase (CDK) inhibitors, enhanced generation of reactive oxygen species (ROS), and aberrant lysosomal activity are several common hallmarks of senescent joint cells, which also release detrimental pro-inflammatory factors known as the senescence-associated secretory phenotype (SASP). Senescence has been implicated in the onset and development of a myriad of age-related diseases, including OA [10,11,12]. Our previous study also revealed that Rhoifolin could alleviate chondrocyte senescence and the OA process via the Nrf2/NF-κB axis [13]. Furthermore, increasing pieces of evidence have shown that mitochondrial dysfunction and associated oxidative stress might induce senescence in joint tissue cells [9,14]. Thus, targeting oxidative stress-mediated senescence is a promising novel treatment that deserves further investigation.

Dendrobine (DEN) is a main bioactive alkaloid isolated from *Dendrobium nobile*, raw plant material of Dendrobii Caulis. It is reported that DEN has effects of anti-inflammation, anti-aging, and neuroprotection [15,16,17]. Recently, emerging research showed that DEN protected H_2_O_2_-induced or UV-induced oxidative stress damage in skin cells or retinal cells [18,19]. However, the effect of DEN on chondrocyte senescence and the OA process has not been studied.

This study aimed to explore the protective effects of DEN on OA both in vitro and in vivo. The role of DEN in the expression of SASP factors was investigated. Chondrocyte senescence, mitochondrial function, intracellular ROS level, and NF-κB pathway were also evaluated in vitro. Also, the role of DEN in alleviating the OA process was assessed in vivo. The present study represented a novel treatment strategy for the treatment of OA.

## 2. Results

### 2.1. The Effect of DEN on the Viability of Chondrocytes

The chemical structure of DEN is shown in Figure 1A. To evaluate the cytotoxic effects of DEN on the chondrocytes, these chondrocytes were treated with different concentrations of DEN (0, 10, 20, 100, 200 μM) for 24 h, and were measured by Cell Counting Kit-8 (CCK-8) assay. Meanwhile, the cytotoxic effects of 20 μM DEN on the chondrocytes during 48 h were also assessed. We found that cell viabilities of rat chondrocytes and C28I2 cells were significantly influenced by DEN only when concentrations were increased to 200 μM (Figure 1B). Meanwhile, during 48 h, 20 μM DEN had no cytotoxic effects on rat chondrocytes and C28I2 cells (Figure 1C).

### 2.2. DEN Inhibited Extracellular Matrix (ECM) Degradation and Increased ECM Synthesis in IL-1β-Treated Chondrocytes

To determine the effect of DEN on IL-1β induced chondrocytes, the expression of matrix-degrading components (matrix metalloproteinase 13 (MMP13) and a disintegrin and metalloproteinase with thrombospondin 5 (ADAMTS5)) and matrix synthesis components (Col2a1 and Aggrecan (ACAN)) was measured. It was shown that the expression of matrix-degrading components was increased and the level of matrix synthesis components was inhibited by IL-1β treatment, whereas the high level of MMP13 and ADAMTS5 and the low level of Col2a1 and ACAN induced by IL-1β were alleviated by DEN treatment, as observed in RT-qPCR and Western blotting (WB) (Figure 2A,B). Immunofluorescence staining also confirmed that DEN inhibited ECM degradation and increased ECM synthesis in chondrocytes induced by IL-1β (Figure 2C and Appendix A).

### 2.3. DEN Inhibited the SASP Factors Expression and Senescence Phenotype in IL-1β-Treated Chondrocytes

To determine the effect of DEN on the expression of SASP factors, the expression of IL6 and TNF-α was detected. RT-qPCR, ELISA, and WB results showed that DEN inhibited SASP factors expression in IL-1β-treated chondrocytes (Figure 3A–C). Further, we investigated the effect of DEN on the senescence phenotype by WB and SA-β-gal assay. The result of WB indicated that protein expressions of p21 and p16 in IL-1β-induced chondrocytes were remarkably higher than those in the control group, while DEN ameliorated this effect (Figure 3C). Meanwhile, SA-β-gal staining showed that DEN could reduce the number of senescent chondrocytes increased by IL-1β (Figure 3D). Taken together, DEN could inhibit cell senescence in IL-1β-treated chondrocytes.

### 2.4. DEN Improved Mitochondrial Function and Reduced the Production of Intracellular ROS in IL-1β-Treated Chondrocytes

Then, we examined the influence of DEN on mitochondrial function and ROS level in rat chondrocytes. We found that the ratio between JC-1 aggregates and JC-1 monomers was remarkably decreased by IL-1β-treatment, and that DEN partially reversed the effect of IL-1β in chondrocytes (Figure 4A,B). This indicates that DEN could increase mitochondrial membrane potential. Meanwhile, the 2′,7′-dichlorodihydrofluorescein diacetate (DCFH-DA) probe demonstrated that DEN could partially reverse the high level of ROS in IL-1β-induced chondrocytes (Figure 4C,D). Overall, these results suggest that DEN improved the IL-1β-induced mitochondrial dysfunction and the high production of intracellular ROS in IL-1β-treated chondrocytes.

### 2.5. DEN Inhibited the ROS/NF-κB Pathway in IL-1β-Treated Chondrocytes

Increasing pieces of evidence have demonstrated that the NF-κB pathway plays a role in accelerating SASP-related cellular senescence [20,21]. Meanwhile, the NF-κB pathway carries out an important role in the OA process [13,22]. Deng et al. [15] have demonstrated that DEN could attenuate osteoclast differentiation through inhibiting the nuclear factor of activated T cells (NFATc1) nuclear translocation. Therefore, we studied whether the NF-κB signaling pathway played a role in DEN’s effect on IL-1β-treated chondrocytes.

Immunofluorescence staining showed that p65 translocated from cytoplasm into the nucleus in chondrocytes treated with IL-1β, and DEN reversed the IL-1β-induced phenomenon (Figure 5A,B). In addition, WB also confirmed that DEN could notably attenuate the degradation of IκB-α, the nuclear translocation of p65, and the phosphorylation of p65 (Figure 5C,D and Appendix A).

Then, we applied NAC (ROS inhibitor) to assess its effects on the NF-κB pathway, ECM, and chondrocyte senescence. We found that NAC further increased the levels of IκB-α and reduced the level of p65, compared with the DEN-treated chondrocytes (Figure 6A). Moreover, WB results showed that NAC significantly increased the expression of Col2a1 and ACAN and reduced the expression of MMP13 and ADAMTS5 in DEN-treated chondrocytes (Figure 6B). Furthermore, the level of IL-6, TNF-a, p21, and p16 was decreased by DEN treatment, and NAC further decreased the level of IL-6, TNF-a, p21, and p16 (Figure 6C). SA-β-gal staining also verified that NAC improved the senescence phenotype compared with the DEN-treated chondrocytes (Figure 6D). Taken together, DEN inhibited the ROS/NF-κB axis in IL-1β-treated chondrocytes.

### 2.6. DEN Alleviates the Progression of OA in an Anterior Cruciate Ligament Transection (ACLT) Rat Model

To detect the influence of DEN on OA in vivo, we constructed an OA model in male rats. After surgery, the rats in OA + DEN group were treated with intragastric administration of 20 mg/kg DEN daily for eight weeks. The results of the X-ray showed that the degree of OA in the OA group was remarkably higher than the sham group, and the degree of OA was significantly decreased in the OA + DEN group compared with the OA group (Figure 7A,B). Similarly, micro-computed tomography (micro-CT) analysis showed that the OA + DEN group presented less osteophyte formation compared to the OA group (Figure 7C,D). Meanwhile, we reconstructed the tibia plateau subchondral bone and analyzed the microstructure parameters. Total bone volume/total tissue volume (BV/TV), trabecular thickness (Tb.Th), and trabecular number (Tb.N) were increased, and trabecular separation (Tb.Sp) was decreased in the OA + DEN group compared with the OA group (Figure 7E,F), which strongly revealed the suppressed bone destruction effect of DEN. Meanwhile, Trap staining showed that DEN might inhibit bone destruction by attenuating osteoclast differentiation (Appendix A).

Further, Safranin O staining and hematoxylin and eosin (HE) staining verified that DEN could significantly attenuate pathological changes, such as ECM loss, superficial cartilage destruction, and synovial inflammation in the ACLT rat model (Figure 8A,C). Meanwhile, the DEN group presented a lower Osteoarthritis Research Society International (OARSI) score and synovitis score compared with the OA group (Figure 8B,D). Furthermore, immunohistochemistry staining indicated that DEN down-regulated the expression of p21, IL6, and MMP13, and up-regulated the expression of Col2a1 in the joints of the OA group (Figure 8E,F). These results indicated that DEN could be used in the treatment of chondrocyte senescence and joint degeneration.

## 3. Discussion

OA has been considered a multifactorial disease rather than a degenerative one, in which low-grade, chronic inflammation plays a crucial role [5,6]. Recently, increasing pieces of evidence show that OA pathology overlaps with the senescence of chondrocytes, and the inhibition of SASP factors might be a promising method to treat OA [23,24]. In this research, we revealed that DEN could reduce ROS levels and alleviate IL-1β-induced SASP in chondrocytes, and delay the progress of OA in the ACLT rat model via the ROS/NF-κB axis, which supported DEN as a promising therapeutic drug for OA.

DEN has been demonstrated to have effects of anti-inflammation and anti-oxidative stress [15,18,19]. In the present study, we revealed that DEN inhibited ECM degradation and increased ECM synthesis. It has been reported that healthy joints maintain articular homeostasis via a balance between chondrocyte anabolic factors and catabolic factors [25]. We proposed that DEN could help to maintain the homeostasis of ECM. Moreover, we found that DEN inhibited the SASP factors expression and senescence phenotype. Overall, our findings showed that DEN was effective in inhibiting inflammation and cell senescence in IL-1β-stimulated chondrocytes.

Mitochondria are significant metabolic centers in chondrocytes, which regulate cellular energy, metabolism, and survival [26,27]. Mitochondrial dysfunction can cause a surge in intracellular ROS level and thus trigger a series of cellular stress events, inducing cell death and degeneration [28,29]. We found mitochondrial dysfunction in IL-1β-induced chondrocytes, including mitochondrial morphology change, reduced mitochondrial membrane potential, and enhanced ROS, while DEN could ameliorate mitochondrial dysfunction and reduce the ROS level. It is reported that ROS through IκB and thioredoxin interacting protein activates the NF-κB and NLRP3 signaling pathway, thereby linking the molecular mechanisms of oxidative stress and inflammatory response [30]. He et al. [31] showed that ROS activated NF-κB and NLRP3 inflammasome pathways to promote inflammation in LPS-induced acute inflammation. Sun et al. [32] demonstrated that ferulic acid could alleviate retinal neovascularization via the ROS/NF-κB axis. The activation of the NF-κB pathway carries out a key role in the progress of OA [33]. Meanwhile, NF-κB pathway regulates the SASP-related gene expression in senescent cells [34]. In our study, we showed that DEN inhibited the NF-κB signaling pathway by reducing the degradation of IκB-α and reversing the p65 translocation to the nucleus.

Further, we evaluated the effects of DEN on OA based on the ACLT rat model. It is well known that OA is characterized by the progressive degradation of articular cartilage, concomitant changes to the subchondral bone, and advancement in osteophytes. In the present study, both X-ray and micro-CT were applied to assess the protective effect of DEN on the development of OA. We found that the OA + DEN group had lower X-ray grades and OASRI scores than the OA group. In addition, the OA + DEN group had higher BV/TV, Tb.Th, and Tb.N, which revealed the suppressed bone destruction effect of DEN. Furthermore, we found that DEN treatment inhibited the SASP factors and increased ECM in the OA model, as indicated by the higher expression of Col2a1 and lower expression of MMP13 in the OA + DEN group. In general, the results above suggest that DEN treatment notably alleviated the progression of OA in rat models.

This study has several potential limitations. Firstly, the protective effect of DEN in OA was only based on the rat model but not on patients with OA. Thus, clinical research on the effects of DEN on OA is still needed. Secondly, DEN was found to reduce intracellular ROS and inhibit NF-κB pathway in IL-1β-induced chondrocytes. However, how the decreased ROS regulates the NF-κB pathway merits further study.

In summary, DEN inhibited SASP factors expression and senescence phenotype in chondrocytes and alleviated the progression of OA via the ROS/NF-κB axis, which provides innovative strategies for the treatment of OA.

## 4. Materials and Methods

### 4.1. Chemicals and Reagents

DEN (molecular formula: C_16_H_25_NO_2_, MW: 263.19, and CAS No. 2115-91-5) was purchased from Chengdu Pufei De Biotechnology Co., Ltd. (Chengdu, China). Recombinant human IL-1β was purchased from PeproTech (Cranbury, NJ, USA). CCK-8 was obtained from Dojindo (Kumamoto, Japan). The cell culture reagents were purchased from Gibco (Grand Island, NY, USA). The primary antibody against GAPDH was obtained from Abcam (Cambridge, MA, USA). The primary antibodies against Col2a1, p21, and ACAN were obtained from Abcolonal (Wuhan, China). The primary antibodies against TNF-α, MMP13, ADAMTS5, p65, β-Actin, and p16^INK4a^ were purchased from Proteintech (Wuhan, China). The primary antibodies against IL-6, IκB-α, and Lamin B were purchased from Abmart (Shanghai, China).

### 4.2. Extraction and Culture of Chondrocytes and Cell Viability Assay

The primary chondrocytes digested from the knees of 4-week-old Wistar rats were plated at a density of 5000 cells per well in 96-well plates in medium (DMEM/F12 medium with 10% fetal bovine serum, 100 mg/mL streptomycin, and 100 U/mL penicillin). In the present study, we used 10 ng/mL IL-1β to construct the in vitro OA model based on previously published studies [35,36,37]. We applied a CCK8 assay to detect cell viability according to the manufacturer’s protocol. The primary chondrocytes were treated with DEN (0, 5, 10, 20, 100, and 200 μM) for 24 h. The absorbance was measured using a microplate reader (Thermo, Rockford, IL, USA) at 450 nm.

### 4.3. Total RNA Extraction and RT-qPCR

Total RNA was isolated from the rat chondrocytes with Trizol reagent following the manufacturer’s protocol. The concentration and purity of the isolated RNA were detected by spectrophotometer and adjusted to 1 μg/μL. The cDNA was amplified using a one-step RT-qPCR reaction. Then, relative mRNA expression was calculated using the 2^−ΔΔCt^ method with GAPDH for normalization. Primers were designed using Primer Premier 5.0, and the rat primer sequences are shown in Table 1.

### 4.4. WB

Chondrocytes were pretreated with DEN and incubated with IL-1β on a 6-well plate. Total intracellular proteins and proteins of the nucleus and cytoplasm were extracted following the instructions. Primary antibodies against IL-6, TNF-α, MMP13, ADAMTS5, Col2a1, ACAN, GAPDH, p65, IκB-α, and Lamin-B were incubated at 4 °C overnight. After being incubated with HRP-conjugated secondary antibody at room temperature for 1 h, the protein bands were measured using an ECL kit on an Imaging System (BIO-ID VL, Conn, France). Image J software was used to visualize optical densities.

### 4.5. SA-β-Gal Staining

Cellular senescence was assessed by detecting the activity of β-galactosidase using an SA-β-gal staining kit (Beyotime) according to the manufacturer’s instructions. Briefly, cultured cells in 6-well plates were immersed in a fixative solution for 15 min at room temperature. After rinsing with PBS, cells were incubated with freshly prepared staining work solution overnight at 37 °C.

### 4.6. Cellular Immunofluorescence Staining

The chondrocytes cultivated were washed three times using ice-cold PBS, fixed in 4% formaldehyde for 15 min, and blocked for 1 h with 3% BSA. The cells were then incubated overnight at 4 °C with primary antibodies, including rabbit anti-Col2a1 (1:200 dilution), anti-MMP13 (1:200 dilution), and anti-p65 (1:200 dilution). The cells were washed and incubated with goat anti-rabbit secondary antibody (1:200 dilution) for 1 h at room temperature. The nuclei were stained using DAPI at 1:500 dilution for 5 min. The slides were washed and fluorescence images were captured using the H550S Photo Imaging System (Nikon, Tokyo, Japan). The staining intensity was determined by measuring the mean optical density in 10 different fields for each sample.

### 4.7. Intracellular ROS

The intracellular ROS level was detected by a ROS assay kit. Firstly, chondrocytes (1 × 10^6^ cells/ well) were seeded in 6-well plates and treated, respectively. Then, the cells were washed with PBS twice and incubated with 10 μM DCFH-DA (Sigma-Aldrich, Shanghai, China) in serum-free medium at 37 °C for 20 min. Then, the chondrocytes were washed with PBS and observed under a confocal microscope (Leica-LCS-SP8-STED, Leica, Germany).

### 4.8. Mitochondrial Membrane Potential

Mitochondrial membrane potential was measured using the JC-1 assay (Beyotime). Cells were observed under a confocal microscope (Leica-LCS-SP8-STED, Leica, Germany). The fluorescence intensity was analyzed by the software ImageJ (NIH, Bethesda, MD, USA), and the ratio of red to green fluorescence reflected the MMP.

### 4.9. Rat Model of OA

Thirty 8-week-old male Wistar rats were purchased from the Experimental Animal Centre of Wuhan University. All animal procedures were performed following the Guidelines for Care and Use of Laboratory Animals of the National Institutes of Health and approved by Experimental Animal Welfare Ethics Committee, Zhongnan Hospital of Wuhan University (ZN2022191, 26 August 2022). After acclimating for 1 week, these male rats were randomly assigned to three groups: sham group, OA group, and OA + DEN group. The OA model was constructed by the transection of anterior cruciate ligament as described [38]. Briefly, after anesthesia with isoflurane, the ACL was transected on the right knee of the rat. After the ACLT surgery, rats in OA plus DEN group were treated with DEN dissolved in sodium carboxymethylcellulose (20 mg/kg/d) by gavage -daily for 8 weeks. Rats in the remaining two groups were given the same volume of sodium carboxymethylcellulose. After eight weeks, knee joint tissues were collected by sacrificing the rats.

### 4.10. X-ray Images

Eight weeks after surgery, X-ray images which were used to quantify the degree of OA were obtained for all animals. The rats were fixed in a prone position in the X-ray irradiation system (Kubtec, Stratford, CT, USA).

### 4.11. Micro-CT

Rat knee joints were harvested and soft tissues including muscles and skins were dissected. The remaining tissues were fixed in 4% formaldehyde. CT scanning was performed using a micro-CT Scanner (Scanco Medical, Bassersdorf, Switzerland) at a resolution of 12 μm/pixel. Three-dimensional reconstruction images were obtained using Scanco Medical software. Osteophytes volume, BV/TV, Tb.N, Tb.Th, and Tb.Sp were analyzed.

### 4.12. Histopathological Analysis

Slides of each cartilage were stained with HE and safranin O-fast green staining. The OARSI guidelines were used to quantify the degeneration of cartilage [39] using a microscope (Nikon, Tokyo, Japan). For immunohistochemical analysis, the sections were placed in a humidified box at 4 °C overnight with primary antibody (Col2a1, MMP13, IL6, and p21, 1:200) and covered. After incubation, the sections were incubated with HRP-conjugated secondary antibodies for 1 h at 37 °C. At least three sections from each specimen were observed. Trap staining was also performed to visualize the effect of selumetinib on osteoclast formation in vivo.

### 4.13. Statistical Analysis

All data were represented by mean ± standard deviation (SD). For data analysis, a Student’s *t*-test was used to evaluate the differences between the two groups. For the three groups, paired *t*-tests and ANOVA were used. A value of *p* < 0.05 was considered statistically different.

## Figures and Tables

**Figure 1 ijms-24-02365-f001:**
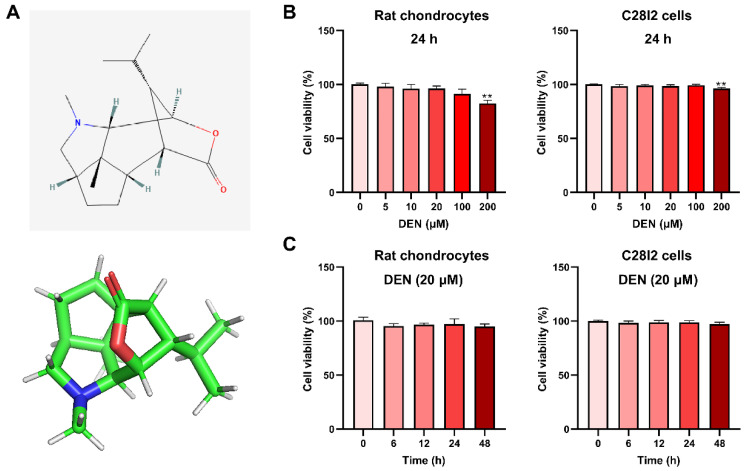
Cytotoxic effects of dendrobine (DEN) on the chondrocytes. (**A**) Chemical structure of DEN. (**B**) The cytotoxic effect of DEN on rat chondrocytes and C28I2 cells was determined at different concentrations (0, 5, 10, 20, 100, and 200 mM) for 24 h using a CCK8 kit. (**C**) The cytotoxic effect of 20 mM DEN on rat chondrocytes and C28I2 cells was determined during 48 h using a CCK8 kit. The data were expressed as mean ± SD of three repeated experiments. ** *p* < 0.01, n = 8.

**Figure 2 ijms-24-02365-f002:**
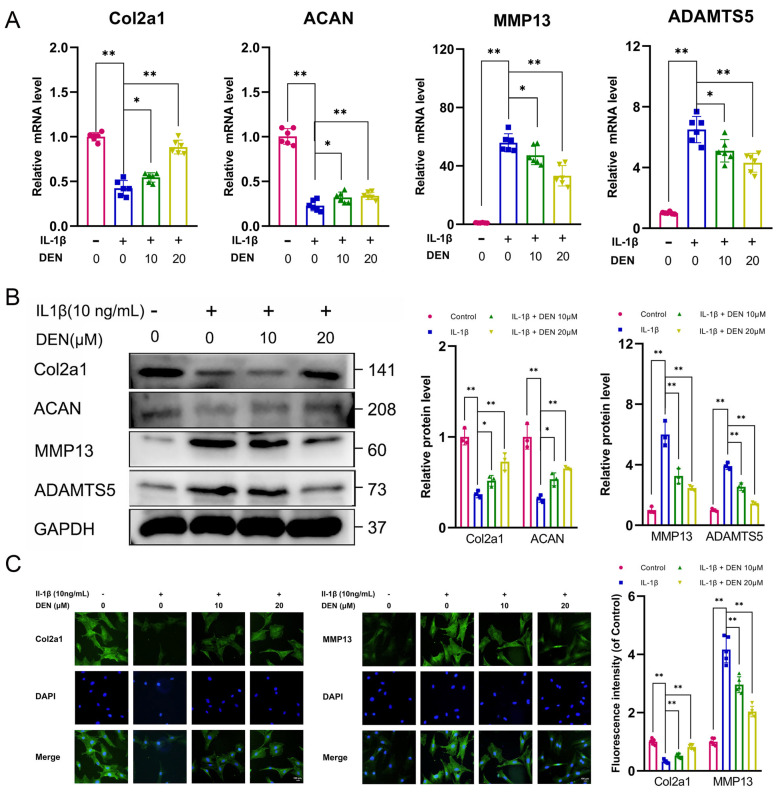
Effects of dendrobine (DEN) on extracellular matrix synthesis and degradation in IL-1β-induced chondrocytes. (**A**) The expression levels of Col2a1, aggrecan (ACAN), MMP13, and ADAMTS5 were shown in the results of RT-qPCR, n = 6. (**B**) The protein levels of Col2a1, ACAN, MMP13, and ADAMTS5 were presented by Western blotting, n = 3. (**C**) The expression of Col2a1 and MMP13 was detected by immunofluorescence using rat chondrocytes. n = 5. The values presented are the means ± SD. * *p* < 0.05, ** *p* < 0.01.

**Figure 3 ijms-24-02365-f003:**
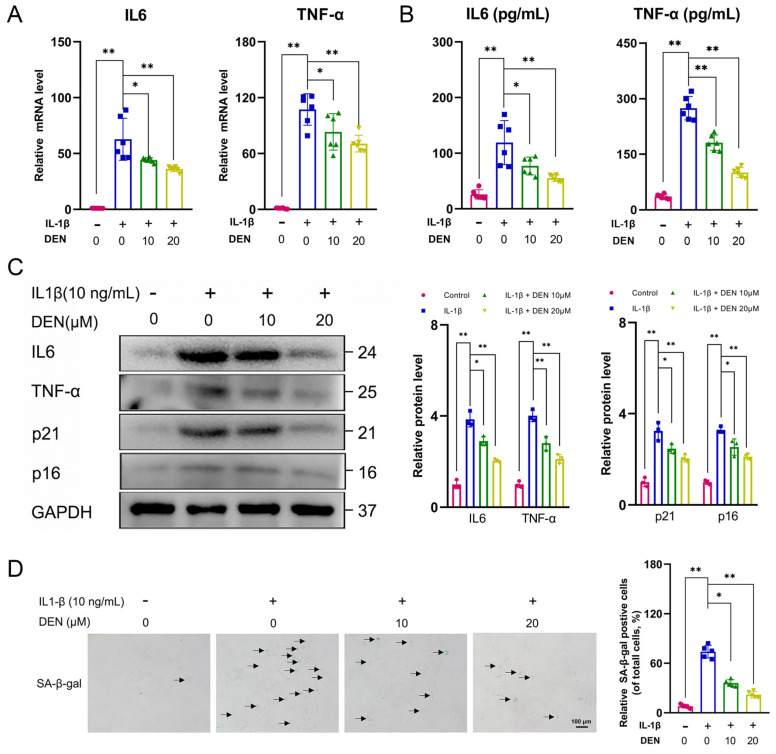
Effect of dendrobine (DEN) on senescence-associated secretory phenotype (SASP) factors expression and senescence phenotype in IL-1β-induced chondrocytes. (**A**) The expression levels of IL6 and TNF-α were shown in the results of RT-qPCR, n = 6. (**B**) The levels of IL6 and TNF-α in the cell supernatants were detected by ELISA, n = 6. (**C**) The protein levels of IL6, TNF-α, p21, and p16 in chondrocytes were presented by Western blotting, n = 3. (**D**) SA-β-gal staining in chondrocytes, n = 5. The data are shown as mean ± SD. * *p* < 0.05, ** *p* < 0.01.

**Figure 4 ijms-24-02365-f004:**
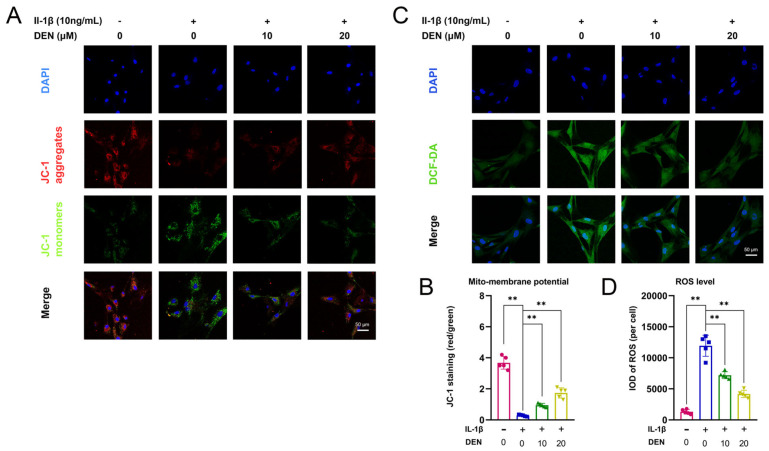
Effects of dendrobine (DEN) on mitochondrial membrane potential and reactive oxygen species (ROS) levels in IL-1β-induced chondrocytes. (**A**,**B**) The mitochondrial membrane potential was measured by JC-1 assay, n = 5. (**C**,**D**) The ROS level in chondrocytes was measured using a ROS assay kit, n = 5. The values presented are the means ± SD. ** *p* < 0.01.

**Figure 5 ijms-24-02365-f005:**
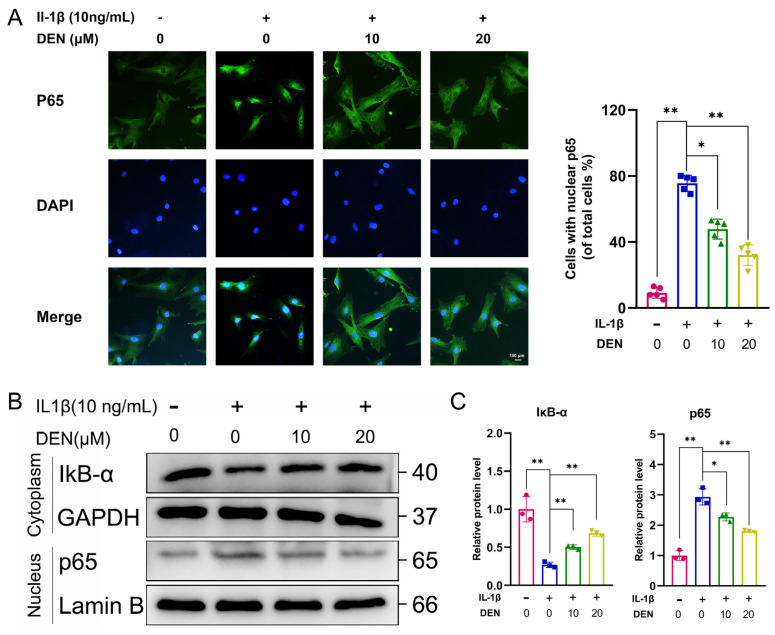
Effects of dendrobine (DEN) on the NF-κB pathway in IL-1β-treated chondrocytes. (**A**) The nuclear translocation of p65 was assessed by immunofluorescence staining, n = 5. (**B**,**C**) The protein levels of p65 in the nucleus and IκB-α in the cytoplasm were presented by Western blotting, n = 3. The values presented are the means ± SD. * *p* < 0.05, ** *p* < 0.01.

**Figure 6 ijms-24-02365-f006:**
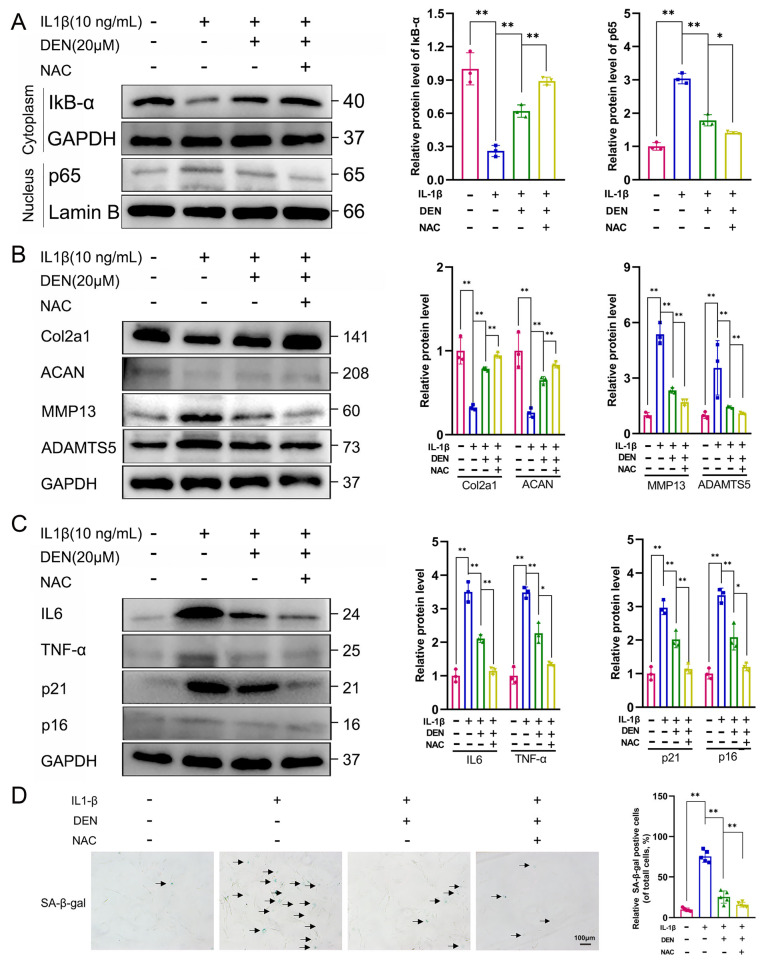
Effect of NAC (ROS inhibitor) on the NF-κB pathway, extracellular matrix, and senescence phenotype in chondrocytes treated by dendrobine (DEN). (**A**) The protein expressions of IκB-α and p65 were assessed by Western blotting in NAC-treated chondrocytes, n = 3. (**B**) The protein expressions of Col2a1, aggrecan (ACAN), MMP13, and ADAMTS5 were assessed by Western blotting in chondrocytes treated with NAC, n = 3. (**C**) The protein expressions of IL6, TNF-α, p21, and p16 were assessed by Western blotting in chondrocytes treated with NAC, n = 3. (**D**) SA-β-gal staining in NAC-treated chondrocytes, n = 5. The values presented are the means ± SD. * *p* < 0.05, ** *p* < 0.01.

**Figure 7 ijms-24-02365-f007:**
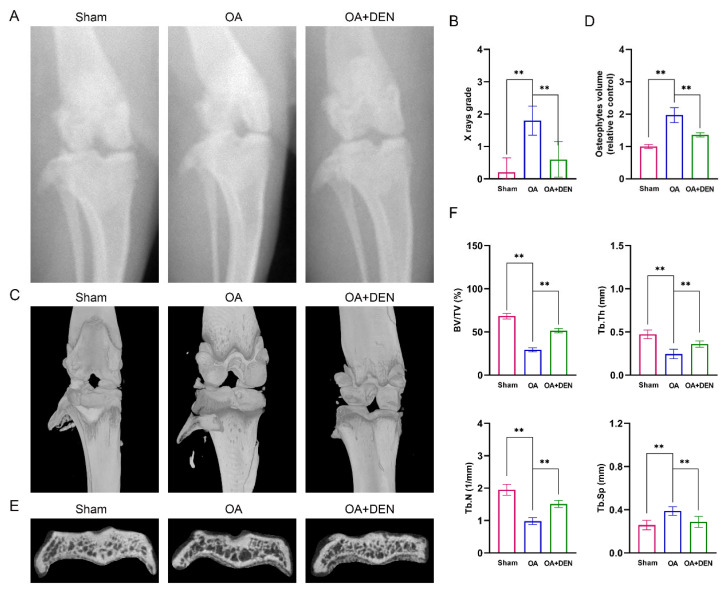
Imaging evaluation of dendrobine (DEN) on the progression of osteoarthritis (OA) in a rat model of OA. (**A**,**B**) X-ray images of rat knee and X-ray grade, n = 5. (**C**,**D**) Micro-CT images of rat knee and osteophytes volume, n = 5. (**E**,**F**) Micro-CT images of subchondral bone and relevant parameters, including total bone volume/total tissue volume (BV/TV), trabecular number (Tb.N), thickness (Tb.Th), and separation (Tb.Sp), n = 5. The values presented are the means ± SD. ** *p* < 0.01.

**Figure 8 ijms-24-02365-f008:**
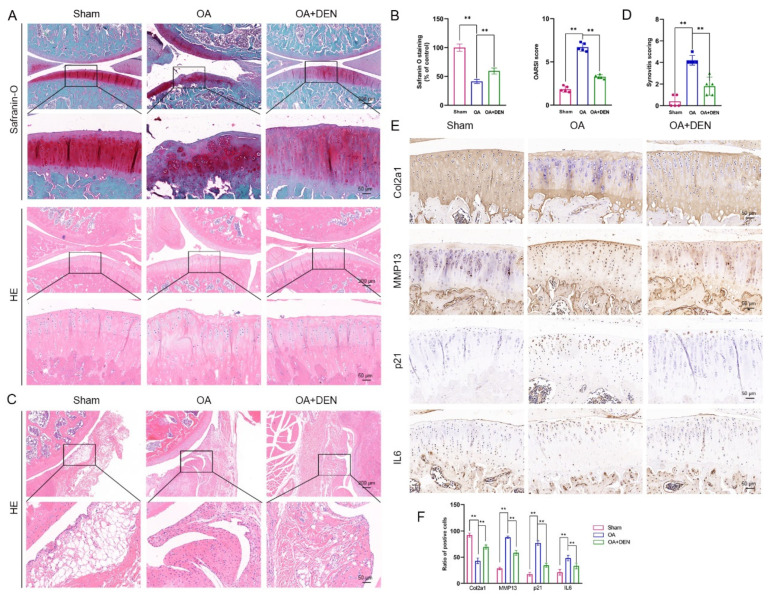
Histological evaluation of dendrobine (DEN) on the progression of osteoarthritis (OA) in a rat model of OA. (**A**,**B**) Morphology of cartilage shown in Safranin O staining and HE staining and the Osteoarthritis Research Society International (OASRI) score, n = 5. (**C**,**D**) HE staining of synovium and synovitis score. (**E**,**F**) Immunohistochemical staining assay of Col2a1, MMP13, p21, and IL6 in the cartilage, n = 3. The values presented are the means ± SD. ** *p* < 0.01.

**Table 1 ijms-24-02365-t001:** Primer used for real-time polymerase chain reaction.

Gene	Forward Primer	Reverse Primer	Annealing
Col2a1	GAGTGGAAGAGCGGAGACTACTG	CTCCATGTTGCAGAAGACTTTCA	60 °C
Aggrecan	CTAGCTGCTTAGCAGGGATAACG	TGACCCGCAGAGTCACAAAG	58 °C
IL-6	GCCAGAGTCATTCAGAGCAAT	CTTGGTCCTTAGCCACTCCT	60 °C
TNF-α	ACCTTATCTACTCCCAGGTTCT	GGCTGACTTTCTCCTGGTATG	60 °C
ADAMTS5	GCAACAAAGTGGGACTACA	GAGAGAATGCATCCCTTAGC	60 °C
MMP-13	GCAGCTCCAAAGGCTACAACTT	GTAATGGCATCAAGGGATAGGG	60 °C
GAPDH	GCAAGTTCAACGGCACAG	GCCAGTAGACTCCACGACA	60 °C

## Data Availability

The data presented in this study are available on request from the corresponding author. The data are not publicly available due to privacy or ethical restrictions.

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
