# Peer review of "Dendrobine Alleviates Cellular Senescence and Osteoarthritis via the ROS/NF-κB Axis"

_ijms, 2023, doi:10.3390/ijms24032365_

Round 1

Reviewer 1 Report

The authors of the study have used a denrobine (DEN) extract to understand whether it modulates senescence associated secretory phenotype (SASP) and reactive oxygen species (ROS). Rat chondrocytes were stimulated with IL-1β and treatment with denrobine resulted in a significant reduction in SASP release that reduced the number of senescent cells. There was also found to be a significant reduction in ROS production that was controlled by the NF-kB pathway. Application of denrobine in a ACL transection model demonstrated that there was a significant improvement in OARSI score with a reduction In MMP13, p21 and IL6 expression.

The results show the promise in using denrobine for treating osteoarthritis. The author needs to consider the following points:-

1)  Why was monolayer culture used for the in vitro experiments rather than pellet cultures ? The latter is an appropriate phenotype for chondrocytes, as in monolayer, they are dedifferentiated.

2)  What was the rationale to use 10ng/mL IL-1β for the stimulation of SASP ? Previous investigations have shown that osteoarthritic tissue has between 0.1-5ng/mL IL-1β.

3) Why were rat chondrocytes used for the data and not human chondrocytes to demonstrate the effectiveness for treating IL-1β treated chondrocytes ? Can the authors present data from human chondrocytes to show its inhibitory properties on IL-1β treated human chondrocytes.

4)  For Figure 3d, can be a better image be produced for the SA-β-gal stained chondrocytes. Same for Figure 6d.

5)    What was the rationale to use an ACL transection model rather than a displaced medial meniscus (DMM) model ?

Author Response

To the reviewer 1:

The results show the promise in using denrobine for treating osteoarthritis. The author needs to consider the following points:

1)  Why was monolayer culture used for the in vitro experiments rather than pellet cultures? The latter is an appropriate phenotype for chondrocytes, as in monolayer, they are dedifferentiated.

Response: Thanks for the reviewer’s comments and warm tips. It is a good choice to use pellet cultures for long-term in vitro experiments, especially for experiments related to cell differentiation. Undoubtedly, pellet culture is an appropriate phenotype for chondrocytes and we’d like to apply this good method in our future research. In our present manuscript, all the in vitro experiments were conducted in three days, which was short-term. We used monolayer culture for the in vitro experiments based on previous published studies [1-5].

  1. Dong, J.; Zhang, K. J.; Li, G. C.; Chen, X. R.; Lin, J. J.; Li, J. W.; Lv, Z. Y.; Deng, Z. Z.; Dai, J.; Cao, W.; Jiang, Q., CDDO-Im ameliorates osteoarthritis and inhibits chondrocyte apoptosis in mice via enhancing Nrf2-dependent autophagy. Acta Pharmacol Sin. 2022, 43 (7), 1793-1802.
  2. Lohberger, B.; Kaltenegger, H.; Eck, N.; Glänzer, D.; Sadoghi, P.; Leithner, A.; Bauer, R.; Kretschmer, N.; Steinecker-Frohnwieser, B., Shikonin Derivatives Inhibit Inflammation Processes and Modulate MAPK Signaling in Human Healthy and Osteoarthritis Chondrocytes. Int J Mol Sci. 2022, 23 (6).
  3. Lu, R.; Yu, X.; Liang, S.; Cheng, P.; Wang, Z.; He, Z. Y.; Lv, Z. T.; Wan, J.; Mo, H.; Zhu, W. T.; Chen, A. M., Physalin A Inhibits MAPK and NF-κB Signal Transduction Through Integrin αVβ3 and Exerts Chondroprotective Effect. Front Pharmacol. 2021, 12, 761922.
  4. Sha, Y.; Zhang, B.; Chen, L.; Wang, C.; Sun, T., Dehydrocorydaline Accelerates Cell Proliferation and Extracellular Matrix Synthesis of TNFα-Treated Human Chondrocytes by Targeting Cox2 through JAK1-STAT3 Signaling Pathway. Int J Mol Sci. 2022, 23 (13).
  5. Teng, Y.; Jin, Z.; Ren, W.; Lu, M.; Hou, M.; Zhou, Q.; Wang, W.; Yang, H.; Zou, J., Theaflavin-3,3'-Digallate Protects Cartilage from Degradation by Modulating Inflammation and Antioxidant Pathways. Oxid Med Cell Longev. 2022, 2022, 3047425.

2)  What was the rationale to use 10ng/mL IL-1β for the stimulation of SASP? Previous investigations have shown that osteoarthritic tissue has between 0.1-5ng/mL IL-1β.

Response: Thanks for the reviewer’s comments. In the present study, we used10ng/mL IL-1β to construct the in vitro OA model based on previous published studies [6-9]. Meanwhile, the SASP factors were detected. We have added the rationale to use 10ng/mL IL-1β in our revised manuscript.

  1. Guo, H.; Yin, W.; Zou, Z.; Zhang, C.; Sun, M.; Min, L.; Yang, L.; Kong, L., Quercitrin alleviates cartilage extracellular matrix degradation and delays ACLT rat osteoarthritis development: An in vivo and in vitro study. J Adv Res. 2021, 28, 255-267.
  2. Wang, F.; Ma, L.; Ding, Y.; He, L.; Chang, M.; Shan, Y.; Siwko, S.; Chen, G.; Liu, Y.; Jin, Y.; Peng, X.; Luo, J., Fatty acid sensing GPCR (GPR84) signaling safeguards cartilage homeostasis and protects against osteoarthritis. Pharmacol Res. 2021, 164, 105406.
  3. Fukui, T.; Yik, J. H. N.; Doyran, B.; Davis, J.; Haudenschild, A. K.; Adamopoulos, I. E.; Han, L.; Haudenschild, D. R., Bromodomain-containing-protein-4 and cyclin-dependent-kinase-9 inhibitors interact synergistically in vitro and combined treatment reduces post-traumatic osteoarthritis severity in mice. Osteoarthritis Cartilage. 2021, 29 (1), 68-77.
  4. Scott, K. M.; Cohen, D. J.; Hays, M.; Nielson, D. W.; Grinstaff, M. W.; Lawson, T. B.; Snyder, B. D.; Boyan, B. D.; Schwartz, Z., Regulation of inflammatory and catabolic responses to IL-1β in rat articular chondrocytes by microRNAs miR-122 and miR-451. Osteoarthritis Cartilage. 2021, 29 (1), 113-123.

3) Why were rat chondrocytes used for the data and not human chondrocytes to demonstrate the effectiveness for treating IL-1β treated chondrocytes? Can the authors present data from human chondrocytes to show its inhibitory properties on IL-1β treated human chondrocytes.

Response: Thanks for the reviewer’s comments. It is more reasonable to use both rat chondrocytes and human chondrocytes for the in vitro experiments. We have added in vitro experiments using human chondrocytes. In the revised manuscript, we evaluated the cytotoxic effects of DEN on rat chondrocytes and C28I2 cells, a human chondrocyte line (Figure 1B-C). Also, we assessed the inhibitory properties of dendrobine (DEN) on IL-1β-treated C28I2 cells (Figure S1).

4)  For Figure 3d, can be a better image be produced for the SA-β-gal-stained chondrocytes. Same for Figure 6d.

Response: Thanks for the reviewer’s comments. We uploaded the wrong picture in our first version. We have re-uploaded the right image in our revised version.

5)    What was the rationale to use an ACL transection model rather than a displaced medial meniscus (DMM) model?

Response: Thanks for the reviewer’s comments. Both an ACL transection model and a displaced medial meniscus (DMM) model were commonly used to construct OA model in rats. We used an ACL transection model by referring to previous literature [4, 10-11].

  1. Sha, Y.; Zhang, B.; Chen, L.; Wang, C.; Sun, T., Dehydrocorydaline Accelerates Cell Proliferation and Extracellular Matrix Synthesis of TNFα-Treated Human Chondrocytes by Targeting Cox2 through JAK1-STAT3 Signaling Pathway. Int J Mol Sci. 2022, 23 (13).
  2. Lin, Y. Y.; Chang, S. L.; Liu, S. C.; Achudhan, D.; Tsai, Y. S.; Lin, S. W.; Chen, Y. L.; Chen, C. C.; Chang, J. W.; Fong, Y. C.; Hu, S. L.; Tang, C. H., Therapeutic Effects of Live Lactobacillus plantarum GKD7 in a Rat Model of Knee Osteoarthritis. Nutrients. 2022, 14 (15).
  3. Zeng, W. N.; Zhang, Y.; Wang, D.; Zeng, Y. P.; Yang, H.; Li, J.; Zhou, C. P.; Liu, J. L.; Yang, Q. J.; Deng, Z. L.; Zhou, Z. K., Intra-articular Injection of Kartogenin-Enhanced Bone Marrow-Derived Mesenchymal Stem Cells in the Treatment of Knee Osteoarthritis in a Rat Model. Am J Sports Med. 2021, 49 (10), 2795-2809.

Thanks again for your valuable comments and constructive suggestions concerning our manuscript. We sincerely wish the revised manuscript could meet the requirement for possible publication. The responses to the editor and all the reviewers were uploaded as an attachment below.

Reviewer 2 Report

This study evaluated the protective role of DEN in articular chondrocytes in the setting of OA.  Overall, the authors examined a broad range of parameters supporting their conclusions.  However, multiple inconsistency were observed in the figures.  Also, the quality of some dataset need to be improved.  Thus, the data in the present version do not fully support the conclusions. 

1, In Fig1, the effect of DEN on cell viability need to be determined in different both primary chondrocytes and cell lines.

2, In Fig2, quantifications of IF stains are required.

3, In Fig3C, better representative WB images of IL-6, TNF, P16, and P21 are required. Looks like the antibodies are not working very well.  In Fig3D, the b-Gal images are not convincing at all, better images are required.  Also, the legend of images indicated NAC was added, however, the quantification showed different doses of DEN. 

4, In Fig4C, the DCF-DA signals in IL-1b-treated cells decreased, however, the quantification showed complete opposite results. 

5, In Fig5, the phosphorylation of p65 need to be evaluated. 

6, In Fig6, again, better representative WB images of IL-6, TNF, P16, and P21 are required.  Also, in Fig6D, the first 3 b-Gal images are the same as in Fig3.  Were these two experiments done at the same time? 

7, In Fig7, the uCT images do not show much osteophyte.  Better representative images are required.  Moreover, the authors conclude the anti-resorptive effect of DEN based on the subchondral bone uCT results.  TRAP staining need to performed to support this conclusion. 

Author Response

To the reviewer 2:

This study evaluated the protective role of DEN in articular chondrocytes in the setting of OA.  Overall, the authors examined a broad range of parameters supporting their conclusions.  However, multiple inconsistency were observed in the figures. Also, the quality of some dataset need to be improved. Thus, the data in the present version do not fully support the conclusions. 

  1. In Fig1, the effect of DEN on cell viability need to be determined in different both primary chondrocytes and cell lines.

Response: Thanks for the reviewer’s comments. We have added the assay of cell viability on C28I2 (a human cell line of chondrocyte) in our revised version (Figure 1B, C).

  1. In Fig2, quantifications of IF stains are required.

Response: Thanks for the reviewer’s advice. We have added the quantifications of IF stains in Figure 2 in our revised version.

  1. In Fig3C, better representative WB images of IL-6, TNF, P16, and P21 are required. Looks like the antibodies are not working very well. In Fig3D, the b-Gal images are not convincing at all, better images are required. Also, the legend of images indicated NAC was added, however, the quantification showed different doses of DEN. 

Response: Thanks for the reviewer’s comments and warm tips. We have replaced the WB images of Il6 and p21 with more representative images. The b-Gal images in Figure 3D were wrongly uploaded in our first version, which contain the effect of NAC. We have replaced Figure 3D with a right and representative image in the revised manuscript. 

  1. In Fig4C, the DCF-DA signals in IL-1b-treated cells decreased, however, the quantification showed complete opposite results. 

Response: Thanks for the reviewer’s comments. The quantification is right in the first version. We have re-uploaded the right image in our revised version.

  1. In Fig5, the phosphorylation of p65 need to be evaluated.

Response: Thanks for the reviewer’s comments. We have evaluated the phosphorylation of p65 in our revised version (Figure S2).

  1. In Fig6, again, better representative WB images of IL-6, TNF, P16, and P21 are required. Also, in Fig6D, the first 3 b-Gal images are the same as in Fig3. Were these two experiments done at the same time? 

Response: Thanks for the reviewer’s comments. We have replaced the WB images of Il6, p21, and p16 with more representative images in our revised version. The images in Figure 3 were wrongly uploaded. We have re-uploaded the right images in our revised version.

  1. In Fig7, the uCT images do not show much osteophyte. Better representative images are required. Moreover, the authors conclude the anti-resorptive effect of DEN based on the subchondral bone uCT results. TRAP staining need to performed to support this conclusion.

Response: Thanks for the reviewer’s comments. We have replaced the uCT images with better representative images in our revised version. Also, we have added TRAP staining in our revised version (Figure S3).

Thanks again for your valuable comments and constructive suggestions concerning our manuscript. We sincerely wish the revised manuscript could meet the requirement for possible publication. The responses to the editor and all the reviewers were uploaded as an attachment below.

Round 2

Reviewer 1 Report

The authors have addressed the points of my review.

Reviewer 2 Report

The authors have addressed all my concerns.